# The Epidemiology of Injuries in Adults in Nepal: Findings from a Hospital-Based Injury Surveillance Study

**DOI:** 10.3390/ijerph182312701

**Published:** 2021-12-02

**Authors:** Santosh Bhatta, Dan Magnus, Julie Mytton, Elisha Joshi, Sumiksha Bhatta, Dhruba Adhikari, Sunil Raja Manandhar, Sunil Kumar Joshi

**Affiliations:** 1School of Health and Social Wellbeing, Faculty of Health and Applied Sciences, University of the West of England, Bristol BS16 1QY, UK; julie.mytton@uwe.ac.uk; 2Centre for Academic Child Health, University of Bristol, Bristol BS8 1NU, UK; dan.magnus@bristol.ac.uk; 3Nepal Injury Research Centre, Department of Community Medicine, Kathmandu Medical College, Kathmandu P.O. Box 21266, Nepal; ejoshi03@gmail.com (E.J.); bhattasumiksha10@gmail.com (S.B.); drsunilkumarjoshi@gmail.com (S.K.J.); 4Mother and Infant Research Activities (MIRA), Kathmandu P.O. Box 921, Nepal; dhrubaadhikari@mira.org.np (D.A.); s.manandhar@mira.org.np (S.R.M.)

**Keywords:** injury surveillance, hospital, program evaluation, developing countries

## Abstract

This study aimed to develop and evaluate a model of hospital-based injury surveillance and describe the epidemiology of injuries in adults. One-year prospective surveillance was conducted in two hospitals in Hetauda, Nepal. Data were collected electronically for patients presenting to emergency departments (EDs) with injuries between April 2019 and March 2020. To evaluate the model’s sustainability, clinical leaders, senior managers, data collectors, and study coordinators were interviewed. The total number of patients with injuries over one year was 10,154, representing 30.7% of all patients visiting the EDs. Of patients with injuries, 7458 (73.4%) were adults aged 18 years and over. Most injuries (6434, 86%) were unintentional, with smaller proportions due to assault (616, 8.2%) and self-harm (408, 5.5%). The median age of adult patients was 33 years (IQR 25–47). Males had twice the rate of ED presentation compared with females (40.4 vs. 20.9/1000). The most common causes were road traffic accidents (32.8%), falls (25.4%), and animal/insect related injuries (20.1%). Most injured patients were discharged after treatment (80%) with 9.1% admitted to hospital, 8.1% transferred to other hospitals, and 2.1% died. In Nepal, hospital-based injury surveillance is feasible, and rich injury data can be obtained by embedding data collectors in EDs.

## 1. Introduction

Injuries are among the leading causes of death and disability in the world, especially in low- and middle-income countries (LMICs), where more than 90% of the world’s injury-related deaths occur [1]. In such settings, preventive health initiatives are limited, and healthcare systems may be poorly prepared to meet this challenge [2,3]. Similar to other LMICs, injuries are a leading cause of morbidity and mortality in Nepal. The estimated proportion of deaths due to injury in Nepal increased from 6.31% to 9.21% between 1990 and 2017 [4]. The injury mortality rate was estimated at 59 per 100,000 population in 2017, doubling the rate calculated from the 2001 Nepal census (30 per 100,000 population) [5]. Young and middle-aged adults experience a greater burden of injuries than other age groups, and there will be more young and middle-aged adults with disabilities secondary to injury in the future [6].

Case series suggest that road traffic injuries (RTIs) are the most common cause of trauma in Nepal [7,8], while falls, burns, poisoning, occupational, and animal-related injuries are other frequent causes [8,9]. The economic and social costs of these injuries can have a serious impact on the individuals, their families, and wider society. In Nepal, the total costs of RTIs alone were estimated at USD 122.88 million, which is equivalent to 1.52% of the gross national product [10]. Injury prevention is relatively new to the public health agenda for Nepal, and the burden of injuries is not well quantified, with most of the literature reporting small hospital case series [11]. Sources of routine injury data collection are also limited [12]. Therefore, there is little evidence to understand the actual burden of injury, the associated risk factors, or the populations at risk.

The World Health Organisation defines injury surveillance as “the ongoing systematic collection, analysis, and interpretation of injury data, for use in planning, implementation, and evaluation of prevention activities [13]”. Hospital-based surveillance systems have been an integral component in improving trauma care in high-income countries [14]. Trauma registries have been successfully established in some LMICs [15,16,17,18]. However, they focus on improving hospital care and outcomes rather than identifying the epidemiology of a condition and prevention opportunities [19,20]. In Nepal, there is currently no formal surveillance system to monitor injury morbidity or mortality. This study aimed to develop and evaluate a hospital-based injury surveillance model in Nepal and describe the epidemiology of adult injuries.

## 2. Materials and Methods

### 2.1. Study Design and Setting

This was a one-year prospective study, conducted in two tertiary care hospitals (Hetauda Hospital and Chure Hill Hospital) in Hetauda, a sub-metropolitan city of Makwanpur district, Nepal. Hetauda is the current provincial headquarters and is located approximately 120 km south-east of Kathmandu. Hetauda Hospital is a government-funded hospital with 110 beds, serving around 300 emergency and outpatient attendances per day. Chure Hill Hospital is a private hospital with 25 beds, serving about 60 emergency and outpatient attendances per day (source: personal communication). Most patients with injuries living locally present to these hospitals due to the long distances to other major hospitals, and limited transportation systems. Both hospitals have the facilities to provide treatment for major and minor trauma [21].

### 2.2. Development and Evaluation of the Surveillance Model

The development and implementation of the surveillance model has been described in detail elsewhere [12]. Briefly, the Emergency Department (ED) leads and hospital management leaders of Hetauda Hospital and Chure Hill Hospital in the Makwanpur district were involved in the development of the injury surveillance model. The model was assessed, modified, and finalised with the engagement and involvement of hospital management committees and local government authorities. A process evaluation was conducted to explore the feasibility of sustaining the injury surveillance system at the two hospital sites and the potential to roll out a similar model to other hospitals.

### 2.3. Data Collection

Trained data collectors recorded anonymised data from injured patients presenting to the EDs of both hospitals, between 1 April 2019 and 24 March 2020, inclusive. Cases included in the study were people who had sustained any new injury presenting to either of the two study hospitals within seven days of the injury event. Injury cases with repeated attendance in the same hospital for the same injury, previous attendance in the other study hospital for the same injury, and hospital attendance for an injury that occurred more than seven days prior to presentation were excluded. The respondent could be the injured person or, if the injured person were unconscious, dead, or under 18 years of age, a family member or caregiver.

To obtain consent and record the information, data collectors approached the patients (or caregivers, where appropriate) once clinical care had been provided. Data were collected using a questionnaire on handheld computers pre-installed with Research Electronic Data Capture (REDCap), Vanderbilt University, Nashville, TN, USA) software [22]. Data included the demographics of the patient and details of the injury event and outcome (date of injury, place of occurrence, activity at the time of injury, mechanism causing the injury, severity of the injury, and disposition). The injury severity was classified as ‘minor’ (superficial injury such as bruises or cuts), ‘moderate’ (injures requiring skilled treatment), or ‘severe’ (injures requiring intensive management), depending on the level of requirement for skilled emergency care intervention. A standardised data collection form was developed from existing tools [13,23,24], adapted for the Nepal context, and piloted prior to data collection. All data collection was conducted in the Nepali language.

For the process evaluation, clinical leads and senior managers at each hospital, plus the data collectors and study coordinators, were invited to participate in face-to-face semi-structured interviews. The interviews were conducted by trained researchers using a topic guide. All of the interviews were audio recorded, transcribed, and translated into English.

### 2.4. Data Analysis

Data analyses were performed using IBM SPSS statistics Windows V.26.0 (IBM Corp. Armonk, NY, USA) [25]. Descriptive statistics were used to express the categorical data in numbers and percentages, while continuous data were described as median and interquartile range. Census data for the Makwanpur population were used to calculate the rates of ED attendance by age and sex. The association between demographic variables and injury severity was assessed using logistic regression and was reported using odds ratios (ORs) and 95% confidence intervals (95% CIs). The ethnicity and castes of patients were grouped into six routinely reported categories, as recommended in the Guidelines for Nepal’s Health Management Information System [26].

The data from the process evaluation were analysed thematically using codes generated iteratively through the reading and re-reading of transcripts [27]. A qualitative data analysis software—NVivo [28]—was used to organise the codes and collate the data relevant to each code. The first 10% of transcripts were double coded to build an agreed coding framework. The coding framework was then applied across all transcripts, and themes relating to sustainability of injury surveillance were identified.

This paper presents the epidemiological information and circumstances surrounding the event of the adult injury cases (18 years and above) and the findings of process evaluation. The epidemiology of injury in children under the age of 18 years is available separately [29].

## 3. Results

### 3.1. Epidemiology of Adult Injuries

Over 12 months, 33,046 patients visited the EDs of the two study hospitals (Figure 1). Of these, 10,154 (30.7%) were patients of any age with injuries, who met the inclusion criteria and agreed to their information being collected. Only 149 patients declined to give consent, 144 were missed, and 15 cases were excluded due to insufficient data being captured. A total of 833 cases were later found not to meet the inclusion criteria. Of patients with injuries, 7458 (73.4%) were adults aged 18 years and older, with 5382 (72.2%) presenting to Hetauda Hospital and 2076 (27.8%) to Chure Hill Hospital. The majority of injuries sustained were unintentional (86.3%), with smaller proportions of assaults (8.2%) and self-harm (5.5%).

#### 3.1.1. Demographics

Among 7458 adults with an injury, males had twice (40.4 per 1000 population) the rate of ED presentations compared to females (20.9 per 1000 population) (Table 1). In males, the rate of injuries decreased with increasing age, but this pattern was not observed in females. Overall, the median age of patients presenting to the ED was 33 years (interquartile range 25–47), with the highest number of attendances among young adults aged 18–44.

#### 3.1.2. Mechanism of Injury by Age and Sex

The mechanisms of injury most commonly reported were road traffic accidents (8.6 per 1000 population), falls (6.6 per 1000 population), and animal/insect-related injuries (5.3 per 1000 population) (Table 2). With regard to intentional harm, self-harm by poisoning and assaults from bodily force (fighting) were the most common causes of injury. Older people were more likely to report falls and animal-related injuries, while road traffic injury was more common in younger patients. Road traffic injuries were the leading cause of injuries among males (13.7 per 1000 population), whereas falls were found to be the most common cause among females (6.2 per 1000 population). Males were three times more likely to be injured in assaults than females (male-to-female ratio: 3.0:1) whilst self-harm by poisoning was more common in females (male-to-female ratio: 0.6:1.0).

#### 3.1.3. Association between Demographics and Injury Severity

A logistic regression model was developed to estimate the association between the demographics of patients with injuries and the severity of injuries (Table 3). Analyses revealed social patterning by sex, age, ethnicity, education, and employment. Men were 35% (95% CI: 20–52%) more likely to have a moderate or severe injury than women. Young adults (18–29 years) had a 25% (95% CI: 3–51%) higher risk of moderate or severe injury than the elderly (≥60 years). The more disadvantaged ethnic groups (Dalit, Janajati, and Madhesi) were 25 to 33% (95% CI: 2–67%) more likely to have a moderate or severe injury compared to those in the more advantaged Brahmin/Chhetri category. Compared to adults with post-secondary education, the risk of moderate or severe injury was 1.7-fold higher in adults without formal education. Unemployed adults were 51% (95% CI: 20–90%) more likely to have a moderate or severe injury than adult students.

#### 3.1.4. Outcomes of Injury

The most common place for an injury to occur was in the home environment (*n* = 3158, 42.4%) followed by the highway/road/street (*n* = 2870, 38.5%), which together accounted for 80.9% of all injuries. The third most common injury location was the workplace, where 1115 (15%) injuries occurred. The remaining injuries took place in recreational areas, schools, or other areas. In terms of injury severity, 4551 (61.1%) were minor or no apparent injury, while 2897 (38.9%) were moderate to severe injuries. Information about injury severity for 10 injured patients was not available.

Disposition data were missing for nine attendees (Figure 2). Most of the injured patients who visited the EDs were discharged after treatment (*n* = 5958, 80%), whilst others were admitted (*n* = 680, 9.1%), transferred to other hospitals (*n* = 603, 8.1%), or left the department against medical advice (*n* = 54, 0.7%). The total number of deaths from injury was 154 (2.1%). Of those, 150 were dead on arrival in the ED and four died after arrival at the hospital. The proportion of unintentional injuries that died was 1.2% (75/6427), while 18.6% (76/408) cases of self-harm died, and 0.5% (3/614) cases of assault.

### 3.2. Findings of the Process Evaluation

Fourteen interviews were completed across the study hospitals. The interviews lasted between 15 and 32 min. Table 4 shows the characteristics of the participants.

Four themes were identified by analysing the data from the 14 interviews: limited number of ED staff and clinical workload; hospitals’ commitment and government support for surveillance; factors that enabled injury data collection; and identified difficulties in collecting injury data. Quotes from the participants are used to illustrate each theme.

**Limited number of ED staff and clinical workload:** Both the government and private hospitals had a shortage of clinical staff in the emergency department. The departments at both hospitals were busy, with a high volume of patients. As a result, the current ED staff were preoccupied with their clinical work and were unable to collect injury data. The collection of injury data was not a hospital requirement for clinical staff. A lack of staff and time constraints in an overburdened emergency care system were identified as major barriers to sustaining the injury surveillance system at the study hospitals.


*Injury data collection work might be difficult for clinical staff in the existing system. They might feel huge pressure if it is being added to their responsibilities. It might be difficult for them to treat the patient and collect the data at the same time.*
(Data collector, female, Hetauda Hospital)


*Many patients visit the emergency department, and sometimes it is extremely busy. Most of the time, only two clinical staff are on duty to provide emergency care. If they [clinical staff] are required to collect data in such a situation, they will not have enough time to do so.*
(Data collector, female, Chure Hill Hospital)

It was suggested that existing ED staff could collect injury data if they were supported and trained to do so, and if the hospital provided them incentives for the extra work. However, the hospital management staff stated that the hospital had limited income sources, particularly in government hospitals, and thus incentives could not come from existing funding. The budget provided by the local government was only intended to support the hospital’s existing staff and healthcare services.


*The government has not provided additional funding for this [injury surveillance] programme. The government only budgets for its own programmes, and each programme has its own staff.*
(Paramedic, male, Hetauda Hospital)

**Hospitals’ commitment and government support for surveillance:** As the study progressed, the clinical staff and hospital management staff began to see the potential value of collecting the injury data. Hospital management committees were also enthusiastic to set up their own injury surveillance system since they recognised the value of surveillance data.


*I really like this programme. This is the first time such a programme has been conducted here. We would be able to learn about the number of injury incidents and the circumstances surrounding them. Similarly, we would know how to improve our emergency care services.*
(ED nurse, female, Chure Hill Hospital)

Injury surveillance, on the other hand, is not currently included in government policy and is not a mandated function in healthcare settings. Hence, sustaining injury surveillance in the absence of government interest and involvement is difficult. Some hospital staff who were aware of the value of the surveillance data expressed interest in the programme continuing, but this was not universal. Participants reported that a strong commitment from the hospital management committees, as well as endorsement and financial assistance from government authorities, would be essential for the long-term viability of the surveillance system in healthcare settings.


*There are ten doctors in this hospital, and only a few of them are familiar with injury surveillance. Others who are uninterested in injury surveillance are unaware of the importance of data and how data collectors are performing their work.*
(Senior doctor, male, Chure Hill Hospital)


*If the surveillance system is a government programme overseen by the Ministry of Health, it must be managed by the hospital. All of the staff will follow the system as if it were their own programme … Injury surveillance in all hospitals is possible if it is led by a higher authority, such as the government or a ministry.*
(Senior doctor, male, Hetauda Hospital)

**Factors that enabled injury data collection:** The hospital management committees and clinical staff at both hospitals supported the surveillance study, and were very helpful to the data collectors and the research team. The data collector supervisor, who had good relationships with the hospital staff, the data collectors, and the research team were particularly important to the success of the study. These observations were supported by the participants in the process evaluation, where the data collectors were regarded as valuable assets to the department. Their clinical backgrounds meant they were used to working in healthcare settings, and had prior experience of approaching patients. This facilitated a positive rapport with the clinical staff in the ED, and created an environment of mutual support.


*Having these data collectors with medical knowledge was extremely beneficial to us. They assist us in the morning and evening, in busy hours, and during emergencies, which is a good thing.*
(Senior doctor, male, Hetauda Hospital)


*Data collection has become simple because the data collectors come from a medical background. It would have been difficult for them if they did not come from a medical background. Because our subject is injury, it is simple for medical background staff to collect data.*
(Study team member, female, MIRA/NIRC)

Every hospital emergency department in Nepal has a police officer on duty. Support from the police officer was particularly helpful for collecting data on road traffic crashes, self-harm, and assault cases, all of which could involve potentially criminal behaviour. In the majority of police cases, the on-duty police officers assisted data collectors not only in approaching the patient but also in obtaining information about the injury.


*We have sought assistance from police officers on occasion. Police officers, like us, are required to record cases such as road traffic injuries. They do not, however, have to collect information as thoroughly as we do.*
(Data collector, female, Chure Hill Hospital)


*In the mortuary case, police introduce our data collectors to the deceased person’s relative and request them to provide information. It has simplified things.*
(ED nurse, male, Hetauda Hospital)

Electronic data collection using handheld computers pre-installed with REDCap made the data collection easier and more efficient.


*I enjoyed collecting data on tablets. Because of the functionality of the REDCap software, it is simple to collect data.*
(Data collector, female, Hetauda Hospital)

**Identified difficulties in collecting injury data:** While we were successful in establishing a surveillance system and collecting injury data over a one-year period, the data collection presented some challenges. The main challenges reported by the participants were collecting the data during busy periods in the ED, in cases that required immediate referral to another hospital, and when there were no relatives or friends to provide information on behalf of severely injured or unconscious patients. A similar difficulty was found when approaching patients who were either drunk or aggressive. Some of the patients were hesitant while answering questions about their caste, educational background, and use of any alcohol or drugs. Data collection with a family member of the deceased was also found to be challenging in the majority of fatal cases, because they were in grief over the loss of a loved one. In such cases, clinical staff provided support to approach family members and supported in the data collection.


*Yes, some of the cases have been missed. It usually happens when a large number of patients present at the emergency department at the same time. Sometimes serious injury cases are immediately referred to another hospital, and we are not aware of these cases because we are busy collecting data on other cases. We only knew about those cases while going through the register.*
(Data collector, female, Hetauda Hospital)


*Collecting data with the relative of a deceased person is quite a difficult task, as they are in shock. Additionally, the relatives of a deceased person leave immediately after post-mortem. They do not wait for us.*
(Study team member, female, MIRA/NIRC)

## 4. Discussion

To our knowledge, this is the first hospital-based injury surveillance study in Nepal, and has shown the considerable burden that injuries place on EDs in this area of Nepal.

In low-income countries such as Nepal, a large proportion of unintentional injuries are caused by road traffic crashes, the causes of which are complex, but include a lack of road safety standards, poor vehicle safety and maintenance, and inadequate implementation of both policies and safe transport infrastructure [30]. These factors are likely to contribute to why road traffic injuries were the leading cause of ED attendance in our study. A similar hospital-based injury surveillance study conducted in India reported that more than half of the patients admitted to the ED had road traffic injuries [17]. In line with the findings of our study, the Indian study also reported that the majority of the victims of road traffic injuries were young men aged 25–44 years. Our data support action to reduce road traffic injuries and suggest that targeting road safety interventions towards young men in Nepal may be appropriate.

Falls were the second leading cause of injury; the finding is consistent with a recent household survey conducted in the same district of Nepal [31]. Case records of injured patients presenting to the ED of one teaching hospital in the Karnali province of Nepal showed that falls were the most common cause of injury [32]. The higher number of fall injuries in this study may have been due to more mountainous terrain with fewer roads than in Makwanpur. This suggests that our findings may only be generalisable to districts in Nepal with similar terrain to Makwanpur (high hills, mid hills, and terai), and not to more mountainous districts.

In Nepal, about 83% of the population is dependent on agriculture and livestock farming for their livelihood, and they reside in rural areas [33]. Many rural families keep domestic animals (e.g., buffalo, goats) and may be injured whilst tending them. Stings from wasps and bees mostly affect working-age farmers in rural Nepal [34]. Snakebites are more common in warmer and wetter months, particularly in terai (plains) regions [35]. Stray dogs are very common in Nepal, and rabies is endemic. About 35,000 animal bites were reported through the Hospital Management Information System in 2017/18, of which 33,000 (94%) were dog bites [36]. These factors may explain why animal/insect-related injuries were the third leading injury cause of attendance in our study, accounting for 20% (1293/6434) of the total unintentional injuries. We found that more than half (52%, *n* = 673) of the animal/insect-related injuries in our study were dog bites. The ready availability of the anti-rabies vaccine at Hetauda Hospital may account for the high number of attendances recorded. Our data suggest the need for public awareness campaigns on the prevention of dog bites and multi-agency rabies prevention programmes.

Poisoning and hanging were the most common mechanisms for self-harm, a finding aligned with other studies included in a recent review of publications reporting injuries in Nepal [11]. Of the 279 cases of self-poisoning, pesticides (67 cases), insecticides (86 cases), and rodenticides (31 cases) were the most commonly used poisoning agents. In terms of fatalities, self-harm accounted for 49.4% (76/154) of all deaths. Of these self-harm deaths, 31.6% (24/76) were caused by poisoning, 65.8% (50/76) by hanging, and 2.6% (2/76) by other mechanisms. Our results support those from an earlier scoping review that reported higher rates of self-harm and suicide among women and younger age groups [37]. Due to the lack of nationally representative suicide incidence data, knowledge of the risk factors for suicide in the Nepalese population is limited [38,39]. Our study shows the lethality of intentional self-harm in this population. Ready access to highly toxic chemicals may mean that those that make an attempt on their life may be more likely to be successful.

Considering all forms of injuries, the risk of sustaining a moderate or severe injury was higher among males, young adults (18–29 years), individuals with low levels of education, and the unemployed. We also found that disadvantaged ethnic groups, such as Dalit, Janajati, and Madhesi, were more likely to have moderate or severe injuries than advantaged ethnic groups such as Brahmin/Chhetri. The predominant ethnic group in the district is Janajati (mostly Tamang) [33], but in the absence of robust population estimates disaggregated by ethnic group, it was not possible to calculate and compare rates of injuries by ethnic groups.

### Strengths and Limitations

The hospitals selected for this study were the two major hospitals in the Makwanpur district, where most injury cases seeking medical assistance were likely to attend. Thus, one of the strengths of this study is the completeness of the sample of patients with injuries studied, as only a small number of patients were likely to attend elsewhere, were missed, or refused to participate. Clinical staff record basic information about patients, such as the reason for attendance and disposition, on a paper-based ED register—no electronic record system is available. In our study, the number of injury cases collected by the data collectors was more than that recorded in the hospital ED registers. This was because the clinical staff were so busy that they did not have time to record all attendances in the register. Therefore, our study provided a more complete record of injuries than previously available, and suggests that retrospective studies relying on ED register data may miss injury cases presenting to hospitals. Collecting data electronically enabled us to monitor the quality of data on a daily basis and provide immediate feedback to the data collectors. Consequently, the data quality was maintained throughout the study period, which improved the validity and utility of the results.

While our study illustrated the burden of injuries presenting to hospitals, it did not capture injury cases treated outside of these two hospitals (such as in primary healthcare settings), or injuries treated by traditional healers or at home. Hospital-based injury surveillance will therefore underestimate the total injury burden. The majority of cases of gender-based violence proceeded directly to a One-Stop Crisis Management Centre on site at Hetauda Hospital and did not present to the ED [40]. Due to the need to maintain the anonymity of these patients, they were not approached to participate in our study. As a result, very few gender-based violence-related injuries appeared in our findings. Our study was required to close one week earlier than planned due to a national lockdown imposed due to the COVID-19 pandemic.

## 5. Conclusions

This study demonstrated that it was feasible to conduct hospital-based injury surveillance in both government and private hospitals in Nepal. The results of this surveillance enabled us to explore injury epidemiology and describe injury inequalities not previously reported in Nepal. Such data can help identify vulnerable populations and can inform the development of targeted preventive measures. These findings may assist hospitals to better understand the type and severity of injury patients they are treating, and may therefore inform training for clinical teams and inform the resources they need. Local government may also find such data valuable when deciding on the local allocation of funding to improve health in the local community, by raising awareness of injury risks or reducing exposure to injury hazards. The process evaluation suggests that rich injury data can be collected, but this is dependent on having dedicated data collectors in Eds, as existing staff do not have the capacity to capture the additional information. This research highlights the potential value of a national-level sentinel injury surveillance system to provide real-time data on injuries in Nepal.

## Figures and Tables

**Figure 1 ijerph-18-12701-f001:**
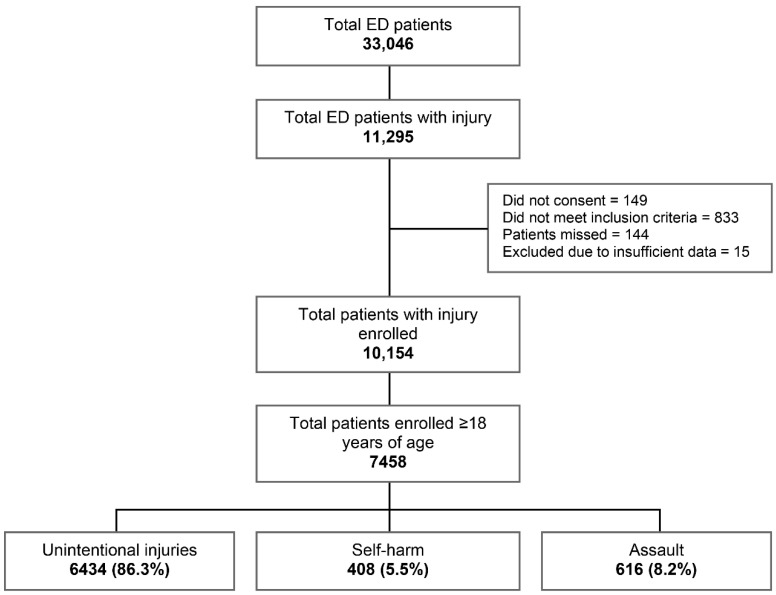
Patients with injuries attending the emergency departments of hospitals in the Makwanpur district, Nepal, April 2019–March 2020.

**Figure 2 ijerph-18-12701-f002:**
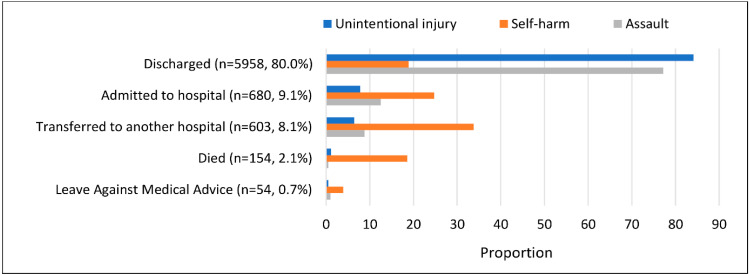
Disposition of injured patients presenting to EDs (*n* = 7449).

**Table 1 ijerph-18-12701-t001:** Number and rates of ED attendance with injury by age and sex.

Age and Sex	Total		Male		Female	
Years	*n*	Rate/1000 (95% CI)	*n*	Rate/1000 (95% CI)	*n*	Rate/1000 (95% CI)
18–29	2950	32.6 (31.5–33.8)	2048	48.6 (46.6–50.7)	902	18.7 (17.5–19.9)
30–44	2378	32.0 (30.8–33.3)	1524	42.8 (40.7–44.9)	854	22.1 (20.6–23.5)
45–59	1285	26.4 (25.0–27.9)	778	31.1 (28.9–33.2)	507	21.5 (19.7–23.4)
60 & above	845	25.7 (24.0–27.4)	459	27.9 (25.4–30.4)	386	23.5 (21.2–25.8)
Total	7458	30.3 (29.6–31.0)	4809	40.4 (39.2–41.5)	2649	20.9 (20.0–21.7)
Median (IRQ *)	33 years (25–47 years)		32 years (24–45 years)		36 years (26–50 years)	

* IQR = interquartile range.

**Table 2 ijerph-18-12701-t002:** Number and rates of EDs attendance with injury-by-injury mechanisms, age, and sex.

Age Groups, Sex Intent, and Mechanisms	18–29 Years	30–44 Years	45–59 Years	≥60 Years	Total	Male	Female	Male-to-Female Ratio
	*n* (rate/1000)	*n* (rate/1000)	*n* (rate/1000)	*n* (rate/1000)	*n* (rate/1000)	*n* (rate/1000)	*n* (rate/1000)	M:F
Unintentional								
Road traffic injury	1005 (11.1)	686 (9.2)	291 (6.0)	130 (4.0)	2112 (8.6)	1628 (13.7)	484 (3.8)	3.4:1
Fall	460 (5.1)	504 (6.8)	339 (7.0)	332 (10.1)	1635 (6.6)	847 (7.1)	788 (6.2)	1.1:1
Animal/insect-related	408 (4.5)	396 (5.3)	288 (5.9)	201 (6.1)	1293 (5.3)	700 (5.9)	593 (4.7)	1.2:1
Stabbed, cut, or pierced	279 (3.1)	215 (2.9)	108 (2.2)	47 (1.4)	649 (2.6)	493 (4.1)	156 (1.2)	3.2:1
Injured by a blunt object	209 (2.3)	157 (2.1)	55 (1.1)	46 (1.4)	467 (1.9)	374 (3.1)	93 (0.7)	4.0:1
Poisoning †	28 (0.3)	23 (0.3)	11 (0.2)	10 (0.3)	72 (0.3)	35 (0.3)	37 (0.3)	0.9:1
Electrocution	35 (0.4)	17 (0.2)	6 (0.1)	5 (0.2)	63 (0.3)	40 (0.3)	23 (0.2)	1.7:1
Fire, burn, or scald	18 (0.2)	16 (0.2)	15 (0.3)	10 (0.3)	59 (0.2)	33 (0.3)	26 (0.2)	1.3:1
Suffocation or choking	5 (0.1)	15 (0.2)	9 (0.2)	4 (0.1)	33 (0.1)	18 (0.2)	15 (0.1)	1.2:1
Other	16 (0.2)	20 (0.3)	10 (0.2)	5 (0.2)	51 (0.2)	28 (0.2)	23 (0.2)	1.2:1
Total	2463 (27.3)	2049 (27.6)	1132 (23.3)	790 (24.0)	6434 (26.1)	4196 (35.2)	2238 (17.6)	1.9:1
Self-harm								
Poisoning ‡	138 (1.5)	82 (1.1)	41 (0.8)	18 (0.5)	279 (1.1)	100 (0.8)	179 (1.4)	0.6:1
Hanging	21 (0.2)	19 (0.3)	13 (0.3)	6 (0.2)	59 (0.2)	32 (0.3)	27 (0.2)	1.2:1
Stabbed, cut, or pierced	40 (0.4)	11 (0.1)	3 (0.1)	0 (0.0)	54 (0.2)	40 (0.3)	14 (0.1)	2.9:1
Other	9 (0.1)	6 (0.1)	1 (0.0)	0 (0.0)	16 (0.1)	12 (0.1)	4 (0.0)	3.0:1
Total	208 (2.3)	118 (1.6)	58 (1.2)	24 (0.7)	408 (1.7)	184 (1.5)	224 (1.8)	0.8:1
Assault								
Bodily force	206 (2.3)	167 (2.2)	69 (1.4)	21(0.6)	463 (1.9)	321 (2.7)	142 (1.1)	2.3:1
Injured by a blunt object	36 (0.4)	27 (0.4)	14 (0.3)	6(0.2)	83 (0.3)	57 (0.5)	26 (0.2)	2.2:1
Stabbed, cut, or pierced	29 (0.3)	14 (0.2)	12 (0.2)	4(0.1)	59 (0.2)	45 (0.4)	14 (0.1)	3.2:1
Other	8 (0.1)	3 (0.0)	0 (0.0)	0(0.0)	11 (0.0)	6 (0.1)	5 (0.0)	1.2:1
Total	279 (3.1)	211 (2.8)	95 (2.0)	31(0.9)	616 (2.5)	429 (3.6)	187 (1.5)	2.3:1

† Of 72 unintentional poisoning: 9 cases from pesticide, 15 from insecticide, 3 from rodenticide, 6 from other medicine, 3 from kerosene, 2 from antiseptic, 19 from wild plants, 10 from other substances, and 5 unknowns. ‡ Of 279 poisonings for self-harm: 67 cases from pesticide, 86 from insecticide, 31 from rodenticide, 5 from sleeping pills, 24 from other medicines, 1 from kerosene, 3 from antiseptic, 1 from wild plants, 18 from other substances, and 43 unknowns.

**Table 3 ijerph-18-12701-t003:** Association between sociodemographic variables and injury severity.

Characteristics	Minor or No Apparent Injury (Total = 4551) *n* (%)	Moderate or Severe Injury (Total = 2897) *n* (%)	Odds Ratio (95% CI)	*p*-Value
Sex				
Male	2886 (60.1)	1915 (39.9)	1.35 (1.20–1.52)	0.000
Female	1665 (62.9)	982 (37.1)	1.00 (reference)	N/A
Age groups (years)				
18–29	1819 (61.7)	1128 (38.3)	1.25 (1.03–1.51)	0.024
30–44	1465 (61.7)	910 (38.3)	1.13 (0.94–1.36)	0.190
45–59	778 (60.6)	506 (39.4)	1.03 (0.86–1.24)	0.730
≥60	489 (58.1)	353 (41.9)	1.00 (reference)	N/A
Ethnicity/caste *				
Dalit	198 (55.8)	157 (44.2)	1.33 (1.06–1.67)	0.014
Janajati	2061 (58.5)	1465 (41.5)	1.25 (1.12–1.39)	0.000
Madhesi	253 (57.9)	184 (42.1)	1.26 (1.02–1.56)	0.031
Muslim	102 (62.2)	62 (37.8)	1.01 (0.73–1.41)	0.935
Brahmin/Chhetri	1845 (65.6)	967 (34.4)	1.00 (reference)	N/A
Others	84 (58.7)	59 (41.3)	1.26 (0.89–1.78)	0.198
Education **				
No formal education	1446 (55.2)	1175 (44.8)	1.70 (1.41–2.05)	0.000
Primary school	1257 (62.0)	772 (38.0)	1.19 (1.00–1.41)	0.048
Secondary school	1065 (65.8)	554 (34.2)	1.03 (0.87–1.22)	0.720
Post-secondary school	781 (67.3)	380 (32.7)	1.00 (reference)	N/A
Occupation ***				
Mainly unemployed	1064 (57.6)	783 (42.4)	1.51 (1.20–1.90)	0.000
Employed salaried	1101 (63.6)	629 (36.4)	1.20 (0.96–1.49)	0.094
Daily wage earners	809 (60.1)	538 (39.9)	1.08 (0.85–1.36)	0.523
Agricultural labourer	681 (59.1)	472 (40.9)	1.23 (0.96–1.58)	0.109
Business owner	467 (62.8)	277 (37.2)	1.19 (0.93–1.53)	0.167
Student	385 (68.6)	176 (31.4)	1.00 (reference)	N/A
Pensioner	44 (75.9)	14 (24.1)	0.69 (0.36–1.33)	0.265

Information about injury severity for 10 injury patients is not available. * Information about ethnicity/caste for 11 injury patients is not available. ** Information about education for 18 injury patients is not available. *** Information about occupation for 8 injury patients is not available. N/A = not applicable.

**Table 4 ijerph-18-12701-t004:** Interview participants’ characteristics.

Participant Number	Interview Participant’s Role in Hospital/Study	Based	Gender	Age (Years)
P1	Data collector	Chure Hill Hospital	Female	20–25
P2	Data collector	Chure Hill Hospital	Female	20–25
P3	Data collector	Hetauda Hospital	Female	35–40
P4	Data collector	Hetauda Hospital	Female	20–25
P5	ED nurse	Chure Hill Hospital	Female	30–35
P6	Senior nurse	Chure Hill Hospital	Female	25–30
P7	Manager	Chure Hill Hospital	Male	25–30
P8	Senior doctor	Chure Hill Hospital	Male	30–35
P9	Manager	Hetauda Hospital	Male	40–45
P10	Senior doctor	Hetauda Hospital	Male	45–50
P11	ED nurse	Hetauda Hospital	Male	40–45
P12	Paramedic	Hetauda Hospital	Male	55–60
P13	Study team member	MIRA/NIRC	Female	40–45
P14	Study team member	MIRA/NIRC	Male	45–50

ED: Emergency Department; MIRA: Mother and Infant Research Activities; NIRC: Nepal Injury Research Centre.

## Data Availability

The datasets used and/or analysed during the current study are available from the corresponding author on reasonable request.

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
