# Peer review of "The Epidemiology of Injuries in Adults in Nepal: Findings from a Hospital-Based Injury Surveillance Study"

_ijerph, 2021, doi:10.3390/ijerph182312701_

Round 1

Reviewer 1 Report

Dear Authors,

Your paper is well written and and present a model of hospital-based injury surveillance and describe the epidemiology of injuries in adults. However, it should be interesting to the readers if you present the model in detail, it was a questionnaire? (if was, present this questionnaire in the paper), it was eletronic?, it was integrated to the health system? (if not how it can be integrate to the national health system).

Thank you for the opportunity to read your paper.

Kind Regards

Author Response

Please find attached our responses to reviewer 1 comments

Reviewer 2 Report

Wonderful study with lots of practical value. The logic and presentation of the study are clear to me. I only have a few minor comments that will help further improve the quality of this study:

1) This study mains focuses on the epidemiology of injuries in adults, while some sentences in this study are discussing the epidemiology of injuries of people under the age of 18 years old (i.e., l.127). I feel like these sentences are irrelevant and should be deleted for the absence of confusion;

2) unclear sentence in l. 165;

3) I understand "-" in Table 3 means the base/reference category, but I would the authors to have a note for this symbol

Author Response

Please find attached our responses to reviewer 2 comments

Reviewer 3 Report

The manuscript presents the results of a study aimed at updating the epidemiology of injuries in adults in Nepal: findings from 2 a hospital-based injury surveillance study

The manuscript is well structured but I believe that the discussion does not sufficiently reinforce the interest of the study itself, i.e. what are the advantages of implementing an injury surveillance model in this context? What measures would be necessary to improve prevention? 

Thank you.

Author Response

Please find attached our responses to reviewer 3 comments
